# Med-SegLens: Latent-Level Model Diffing for Interpretable Medical Image Segmentation

**Salma J. Ahmed** [1]   **Emad A. Mohammed** [1]   **Azam Asilian Bidgoli** [1]

## Abstract

Modern segmentation models achieve strong predictive performance but remain largely opaque, limiting our ability to diagnose failures, understand dataset shift, or intervene in a principled manner. We introduce **Med-SegLens**, a model-diffing framework that decomposes segmentation model activations into interpretable latent features using sparse autoencoders trained on SegFormer and U-Net. Through cross-architecture and cross-dataset latent alignment across healthy, adult, pediatric, and sub-Saharan African glioma cohorts, we identify a stable backbone of shared representations, while dataset shift is driven by differential reliance on population-specific latents. We show that these latents act as causal bottlenecks for segmentation failures, and that targeted latent-level interventions can correct errors and improve cross-dataset adaption without retraining, recovering performance in 70% of failure cases and improving Dice score from 39.4% to 74.2%. Our results demonstrate that latent-level model diffing provides a practical and mechanistic tool for diagnosing failures and mitigating dataset shift in segmentation models. ⌗

## 1. Introduction

Medical image segmentation underpins computer-aided diagnosis, treatment planning, and disease monitoring (Pham et al., 2000). Despite strong benchmark performance from convolutional and transformer-based models (Sultana et al., 2020; Xiao et al., 2023; Ahmed et al., 2025), their internal representations remain largely opaque. This limits our ability to diagnose systematic failures, understand dataset shifts, or intervene in a principled way, particularly in high-

stakes clinical settings where models are deployed across heterogeneous populations (Holzinger et al., 2017).

Current mitigation strategies rely on retraining or post hoc interpretability methods (Selvaraju et al., 2017; Schlemper et al., 2019). While useful for visualization, these approaches provide limited causal insight into how internal representations encode population-specific priors or drive errors under distribution shift (Adebayo et al., 2018; Rudin, 2019). As a result, failures are difficult to attribute, compare across datasets, or correct without costly retraining.

We introduce **Med-SegLens**, a mechanistic *model-diffing* framework for medical image segmentation that analyzes how internal representations differ across datasets Figure 1. Med-SegLens applies sparse autoencoders (SAEs) to intermediate activations of segmentation models, enabling a sparse, latent-level decomposition of learned features (Templeton et al., 2024; Gao et al., 2024). To ground these latents in the imaging domain, we propose a geometry- and spatially grounded automated interpretation pipeline that assigns anatomical and pathological semantics.

Our central contribution is a latent alignment and diffing procedure that decomposes representation shifts into *shared* and *dataset-specific* latent features. We show that dataset-specific latents act as causal bottlenecks for segmentation performance: intervening on them through targeted ablation or steering induces predictable changes in model behavior. Across multiple cohorts and architectures (SegFormer (Xie et al., 2021) and U-Net (Ronneberger et al., 2015)), this enables diagnosis and partial mitigation of segmentation failures *without retraining*, recovering performance in 70% of failure cases and improving Dice on the most affected failure class from 39.4% to 74.2%.

Our work suggests that population-specific priors in medical segmentation models are encoded in identifiable and manipulable latent features. Med-SegLens offers a principled framework for analyzing dataset shift and exploring representation-level interventions for segmentation failures. Our contributions are summarized as follows:

- We introduce **Med-SegLens**, the *first* mechanistic *model-diffing* framework for medical image segmentation, enabling principled comparison of segmentation

---

[1]Department of Computer Science and Physics, Wilfrid Laurier University, Waterloo, Canada. Correspondence to: Salma J. Ahmed <ahme3460@mylaurier.ca>.

*Proceedings of the 43rd International Conference on Machine Learning*, Seoul, South Korea. PMLR 306, 2026. Copyright 2026 by the author(s).

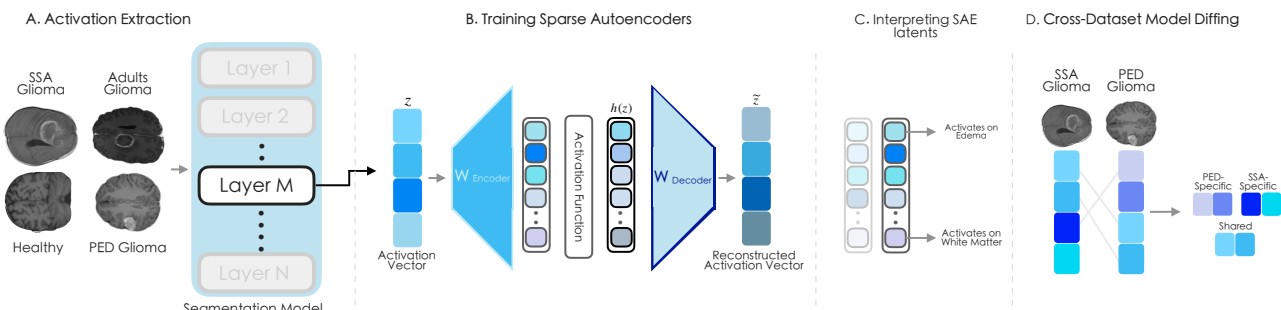

*Figure 1.* Overview of the proposed framework. We extract intermediate activations from a medical image segmentation model (A) and train sparse autoencoders on these activations to learn interpretable latent features (B). Using our Auto-Interp method, we assign semantic meaning to the learned latents (C). Finally, we perform cross-dataset model diffing to characterize domain shift and enable targeted latent-level interventions (D).

models across datasets via interpretable representations.

- We propose a geometry- and spatially grounded automated method for interpreting latent features.

- We show that cross-dataset model diffing reveals shared and population-specific latents, with the latter acting as causal bottlenecks under dataset shift.

- We demonstrate that segmentation failures can be traced to individual latents and mitigated through targeted interventions, recovering performance in 70% of failure cases.

- Without retraining, we improve cross-dataset adaptation by mitigating class-specific bias, increasing Dice from 39.4% to 74.2% on the most affected class.

## 2. Related Work

Our work bridges mechanistic interpretability, model diffing, and medical image segmentation by using Sparse autoencoders to extract interpretable latent representations and the Hungarian algorithm to analyze cross-dataset representation shift.

**Interpretability in Medical Image Segmentation** Most interpretability approaches for medical image segmentation rely on post hoc saliency or attention-based techniques, such as SHAP and Grad-CAM and its variants (Selvaraju et al., 2017; Lundberg & Lee, 2017; Schlemper et al., 2019). While these methods highlight regions correlated with predictions, they do not reveal the internal representations driving segmentation decisions, explain failures under dataset shift, or enable principled error diagnosis or control (Rudin, 2019).

**Mechanistic Interpretability** aims to explain neural network behavior by reverse-engineering models to identify the algorithms, features, and circuits they implement (Olah

et al., 2017; Elhage et al., 2022). Recent work has made substantial progress in identifying task-relevant circuits in language models (Nanda et al., 2023; Olah et al., 2020; Cunningham et al., 2023) and in vision models, primarily for classification backbones such as ViTs/CLIP (Dosovitskiy, 2020; Achtibat et al., 2022; Gandelsman et al., 2024; 2023; Komorowski et al., 2023; Zaigrajew et al., 2025). However, analogous mechanistic analyses for dense prediction tasks such as medical image segmentation remain largely unexplored. We extend mechanistic interpretability to segmentation by uncovering how models represent anatomical structure, why failures arise under dataset shift, and how targeted latent interventions can correct these failures.

**Model Diffing and Dataset Shift** aims to identify changes in the internal representations of two models. Methods such as CrossCoders (Lindsey et al., 2024) have been used to analyze representation differences in language models, including risks and fine-tuning side effects (Kassem et al., 2025; Minder et al., 2025; Boughorbel et al., 2025; Wang et al., 2025; Jiralerspong & Bricken). The Hungarian algorithm has also been proposed for model alignment (Paulo & Belrose, 2025). However, applying model diffing to study dataset shift or cross-dataset representation learning remains largely underexplored in segmentation models. We apply model diffing to models trained on different datasets to characterize representation shift and adaptation.

## 3. Background and Methods

### 3.1. Problem Setting: Representation-Level Model Diffing

Our goal is to understand how segmentation models differ internally when trained on distinct medical populations, and how these differences relate to systematic failures under dataset shift. Rather than analyzing a single model in isolation, we adopt a *model diffing* perspective: given two segmentation models trained on different datasets but sharing the same architecture, we explicitly compare their

internal representations to identify (i) shared, population-invariant features and (ii) dataset-specific features that act as causal bottlenecks for performance (Algortihm 1).

Formally, let $f_{\theta(d)} : \mathcal{X} \to \mathcal{Y}$ denote a segmentation model trained on dataset $d$. Given two datasets $d_1, d_2$, our objective is to characterize differences in the internal feature spaces of $f_{\theta(d_1)}$ and $f_{\theta(d_2)}$ at the level of interpretable latent components, rather than raw activations or output predictions.

### 3.2. Population Knowledge and Datasets

**Datasets.** We study representation shift across brain MRI datasets spanning distinct populations and pathologies. Healthy anatomy is represented by the IXI dataset (Consortium, 2012), while pathological domains are drawn from the BraTS 2023 challenge (Menze et al., 2014; Baid et al., 2021), including adult glioma, pediatric glioma, and sub-Saharan African (SSA) glioma cohorts. We use a preprocessed version of IXI (Chen et al., 2022) with voxel-level subcortical segmentation masks, mapping the original anatomical labels to a reduced set of target classes.

Although all datasets consist of T1-weighted brain MRIs, they differ substantially in population demographics, disease presentation, and acquisition distributions, making them well-suited for studying population-specific versus shared representations. Additional statistical details are provided in Appendix A.

### 3.3. Controlled Model Training

To isolate representation differences due solely to data, we train the *same segmentation architecture independently* on each dataset. We experiment with SegFormer-B4, a Transformer-based model, and U-Net as a convolutional baseline. With four tumor classes for BraTS cohorts and nine anatomical classes for IXI.

All input images are resized to $256 \times 256$. Training uses standard data augmentation, while validation uses deterministic preprocessing. This results in four dataset-specific models:

$$\{ f_{\theta(\text{Adult})}, \ f_{\theta(\text{PED})}, \ f_{\theta(\text{SSA})}, \ f_{\theta(\text{IXI})} \},$$

which form the basis for all subsequent model-diffing analyses.

### 3.4. Latent Extraction with Sparse Autoencoders

Directly diffing raw activations is challenging due to their high dimensionality and lack of semantic structure. To obtain a disentangled and comparable representation space, we decompose internal activations using sparse autoencoders (SAEs).

For each trained segmentation model, we extract activations

**Algorithm 1** Latent-Level Model Diffing for Segmentation

---

**Require:** Datasets $\{\mathcal{D}_m\}_{m=1}^{M}$, architecture $f_\theta$, sparsity $k$, threshold $\tau$
**Ensure:** Shared and dataset-specific latent features
1: **for** each dataset $\mathcal{D}_m$ **do**
2:      Train segmentation model $f_{\theta(m)}$ on $\mathcal{D}_m$
3:      Extract intermediate activations $X^{(m)}$
4:      Train BatchTopK SAE on $X^{(m)}$
5:      Obtain encoder and decoder weights
6: **end for**
7: **for** each dataset pair $(m, n)$ **do**
8:      **for** $p \in \{\text{enc}, \text{dec}\}$ **do**
9:          Compute cosine similarity matrix $C^p$
10:          Compute matching $\pi^p$ using Hungarian algorithm
11:      **end for**
12:      Identify shared latent indices satisfying:
13:      $\pi^{\text{enc}}(i) = \pi^{\text{dec}}(i)$
14:      $C_{i,\pi(i)}^{\text{enc}} \geq \tau$ and $C_{i,\pi(i)}^{\text{dec}} \geq \tau$
15: **end for**

---

from a fixed intermediate layer[1] and train a BatchTopK sparse autoencoder (Bussmann et al., 2024) (Figure 1A-B). Given a batch of activations $X \in \mathbb{R}^{n \times d}$, the SAE computes

$$Z = \text{BatchTopK}(XW_{\text{enc}} + b_{\text{enc}}), \qquad \hat{X} = ZW_{\text{dec}} + b_{\text{dec}},$$

where $Z \in \mathbb{R}^{n \times m}$ is a sparse latent representation. The BatchTopK operator retains the top $n \times k$ activations across the batch, enforcing an average sparsity of $k$ active latents per sample while allowing adaptive allocation based on sample complexity.

The SAE is trained using reconstruction loss

$$\mathcal{L}_{\text{SAE}} = \|X - \hat{X}\|_2^2.$$

We set $k = 32$ and use an expansion factor of 16. Smaller expansion factors produced broader, less interpretable features, while larger expansions encouraged feature splitting and yielded more monosemantic latents (Bricken et al., 2023) (subsection B.2). Each dataset thus yields an SAE whose latent dimensions serve as the atomic units for model diffing.

### 3.5. Automated Latent Semantics Discovery

To interpret SAE latents, we analyze their Top-$K$ highest-activating samples and corresponding spatial activation maps (Figure 1C). For each latent $z_i$, we generate activation heatmaps by projecting latent activations back to the spatial resolution of the segmentation model and overlay them on the input image and ground-truth mask.

---

[1]We select a middle layer, as prior work shows it captures the richest semantic structure (Skean et al., 2025) Appendix B.

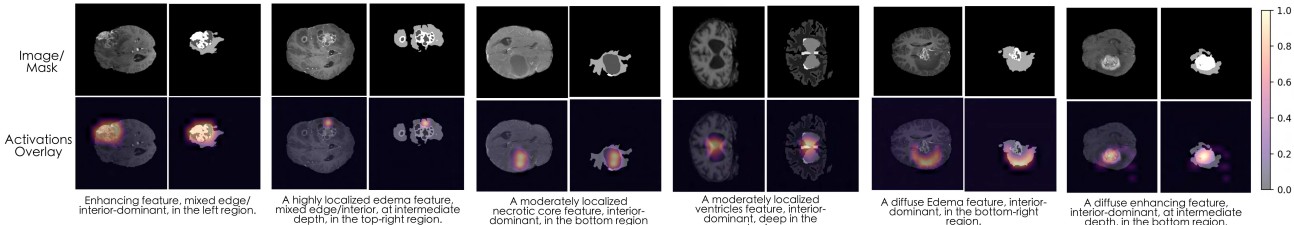

*Figure 2.* Examples of SAE features from SegFormer and U-Net. Representative SAE latents capture tumor subregions, including edema, enhancing tumor, and necrotic core, as well as healthy anatomy, with diverse spatial and morphological patterns. Columns 1, 3, 4, and 5 correspond to SegFormer features. Each latent is shown with its auto-interpreted description; the top row shows the MRI image and ground-truth mask, while the bottom row overlays the SAE activation heatmap on the MRI and segmentation mask.

We further introduce an automated interpretation pipeline that maps latent activations to structured anatomical semantics. For each latent, we compute geometry- and spatial-based metrics, including brain-edge ratio, depth within brain tissue, spatial entropy, and centroid localization. These metrics provide quantitative descriptors of *where* and *how* a latent activates (Further Details Appendix C).

Latents are interpreted in a dataset-specific context: for BraTS, we assess alignment with tumor subregions (edema, enhancing tumor, necrotic core), while for IXI we analyze correspondence with healthy anatomical structures such as white matter, gray matter, and ventricles. This step grounds latent features in clinically meaningful concepts and enables principled comparison across datasets.

### 3.6. Cross-Dataset Model Diffing via Latent Alignment

Our core contribution is a latent-level model diffing procedure that explicitly aligns sparse autoencoder (SAE) features across datasets. Because SAE latent indices are arbitrary, direct comparison requires solving a feature correspondence problem (Figure 1D).

Given two SAEs trained on datasets $d_1$ and $d_2$, with encoder and decoder weights $W_{\text{enc}}^{(1)}, W_{\text{dec}}^{(1)}$ and $W_{\text{enc}}^{(2)}, W_{\text{dec}}^{(2)}$, we compute cosine similarity matrices between corresponding latent vectors:

$$C_{ij}^p = \cos\left(W_{p,i}^{(1)}, W_{p,j}^{(2)}\right), \quad p \in \{\text{enc}, \text{dec}\}.$$

To obtain a strict one-to-one correspondence between latents, we apply the Hungarian algorithm to each similarity matrix, yielding matchings

$$\pi^p = \text{Hungarian}(C^p), \quad p \in \{\text{enc}, \text{dec}\}.$$

We define the set of *shared* latents as

$$\mathcal{S} = \left\{ i \mid \pi^{\text{enc}}(i) = \pi^{\text{dec}}(i), \ C_{i,\pi(i)}^{\text{enc}} \geq \tau, \ C_{i,\pi(i)}^{\text{dec}} \geq \tau \right\},$$

where $\tau$ is a similarity threshold. We set $\tau = 0.8$, selected via a sweep as the value that best balances stability and coverage. Latents outside $\mathcal{S}$ are classified as dataset-specific.

This conservative criterion isolates a stable backbone of population-invariant representations while identifying dataset-specific latents that differ systematically across models. The resulting alignment underpins our downstream analyses of representation shift, failure diagnosis, and latent intervention.

## 4. Latent Feature Analysis

### 4.1. SAEs Uncover Interpretable Features

Our automated interpretation framework reveals that sparse autoencoders decompose segmentation model activations into a set of semantically meaningful latent features. These latents capture both generic visual patterns, such as background structure and edge-related information, as well as domain-specific concepts corresponding to tumor subregions and healthy anatomical structures.

Figure 2 shows representative SAE latents and activation heatmaps from both U-Net and SegFormer. Together with automatically generated semantic labels, these results demonstrate that individual latents align with distinct visual and anatomical concepts, consistently across architectures and datasets. Rather than a single latent per class, multiple latents emerge for each tumor or anatomical category, capturing variation in spatial location, morphology (e.g., diffuse vs. localized), scale, and depth within the brain (Appendix D). This decomposition shows that SAEs disentangle fine-grained, clinically relevant structure, encoding tumors not only by presence but by diverse, appearance-dependent features.

For evaluation beyond brain MRI on a CT multi-organ dataset and a non-medical dataset, see Appendix F.

### 4.2. Cross-Architecture Internal Reasoning

We compare how SegFormer and U-Net utilize internal representations by analyzing SAE latent activations under identical training and inference conditions. Using 100 BraTS-Adult cases, we aggregate activation mass per latent and group features into tumor, background, and boundary categories.

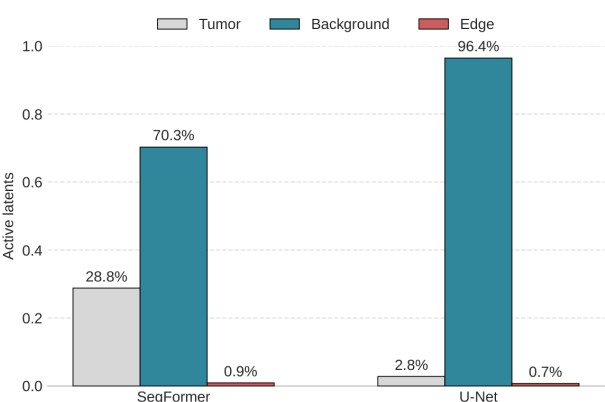

*Figure 3.* Distribution of active SAE latents across semantic categories for SegFormer and UNet on 100 BraTS-Adult cases, grouped by automated semantic interpretation.

As shown in Figure 3, both models are dominated by background features, reflecting the prevalence of non-tumor anatomy in MRI. However, latent usage differs sharply: U-Net relies almost exclusively on background latents (Tumor: 2.8%), while SegFormer activates substantially more tumor-related latents (Tumor: 28.8%), yielding a more balanced representation and highlighting clear architectural differences in internal feature utilization.

## 5. Do Models Learn Shared Representations Across Datasets?

A central question in medical image segmentation is whether models trained on different populations learn shared internal representations or diverge under dataset shift. Although all datasets depict the same organ (the brain), they differ substantially in acquisition protocols, demographics, and disease prevalence.

We study four domains healthy adults, adult glioma, pediatric glioma (PED), and Sub-Saharan African (SSA) glioma and train the same architecture independently on each using matched optimization settings, isolating the effect of data distribution on learned representations.

### 5.1. Model Diffing via Sparse Autoencoders

We analyze representation differences across datasets using our *latent-level model diffing framework* based on sparse autoencoders (SAEs). SAEs provide an interpretable latent basis that enables direct, feature-level comparison between independently trained segmentation models.

### 5.2. Shared and Dataset-Specific Representations

Table 1 reports the fraction of shared latents between dataset pairs. All pairs show non-zero overlap, indicating a partially shared internal basis across populations. Adult–pediatric models share the most latents, followed by adult–SSA

*Table 1.* Fraction of shared SAE latents of SegFormer trained on different dataset cohorts.

| Dataset Pair | Shared Latents (%) |
|---|---|
| Adult Glioma ↔ Pediatric Glioma | 58.9 |
| Adult Glioma ↔ Sub-Saharan Glioma | 43.2 |
| Adult Glioma ↔ Healthy Brains | 36.1 |

and adult–healthy (see Appendix E for additional pairs). Shared representations are more prevalent among diseased cohorts than between diseased and healthy data, consistent with tumor-specific structure. The lower overlap with SSA likely reflects domain shift from scanner and image-quality differences (Adewole et al., 2023). Overall, these results show that segmentation models learn both population-invariant and dataset-specific representations, explaining cross-dataset generalization and failures under distribution shift.

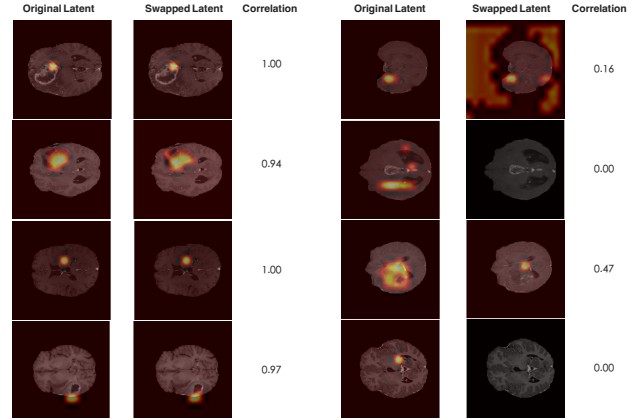

*Figure 4.* Feature swapping between Adult and Pediatric SAEs. Shared latents (left) preserve activation structure and spatial correlation, while non-shared latents (right) produce disrupted or absent activations.

### 5.3. Causal Role of Shared Latents

We test the causal role of shared latents via *feature swapping* between SAEs trained on Adult and Pediatric data. Let $h \in \mathbb{R}^d$ denote activations from a fixed model layer. An SAE trained on dataset $D \in \{\text{Adult}, \text{Pediatric}\}$ encodes and reconstructs

$$z^{(D)} = W_{\text{enc}}^{(D)} h, \qquad \hat{h}^{(D)} = W_{\text{dec}}^{(D)} z^{(D)}.$$

Given aligned latent indices $\mathcal{I}$, we swap shared components:

$$\tilde{z}_i^{(\text{Adult})} = \begin{cases} z_i^{(\text{Pediatric})}, & i \in \mathcal{I}, \\ z_i^{(\text{Adult})}, & \text{otherwise.} \end{cases}$$

The modified activation is

$$\tilde{h} = W_{\text{dec}}^{(\text{Adult})} \tilde{z}^{(\text{Adult})},$$

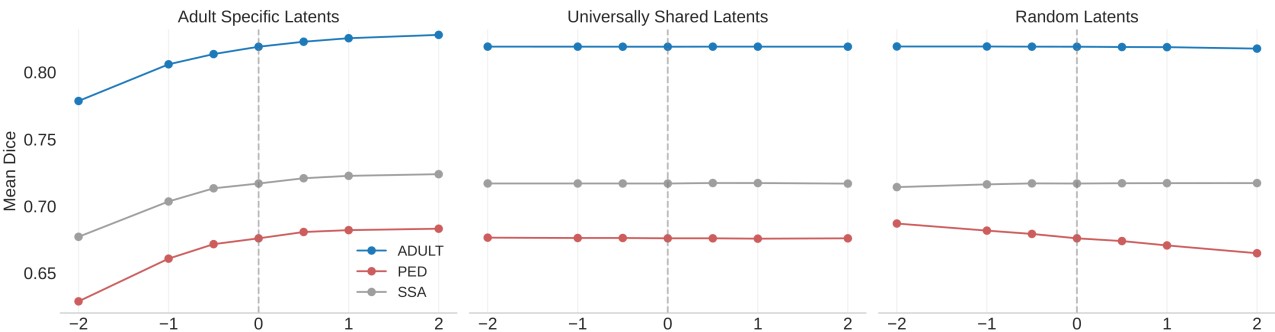

*Figure 5.* Mean Dice versus steering strength ($\alpha$) for adult-specific (left), universally shared (middle), and random (right) SAE latents. Adult-specific steering strongly affects Adult performance and weakly affects Pediatric and SSA, while shared or random steering yields minimal, non-selective changes.

which is propagated through the remaining network with all weights fixed.

As shown in Figure 4, swapping shared latents preserves spatial localization and semantic structure, with high correlation to original activations, indicating domain-invariant representations. In contrast, swapping non-shared latents yields diffuse or inactive responses and low correlation, reflecting domain-specific features. These results provide causal evidence that encoder–decoder aligned latents act as transferable mechanisms, while non-aligned latents do not (see Appendix E).

### 5.4. Universally Shared Latents

Beyond pairwise overlap, we examine whether *universally shared* latents exist—features identified as shared across all dataset pairs. Let $\mathcal{S}_{m,n}$ denote shared latents between datasets $m$ and $n$; universally shared latents are

$$\mathcal{S}_{\text{univ}} = \bigcap_{m \neq n} \mathcal{S}_{m,n}.$$

All remaining latents are treated as population-specific.

To assess their role, we steer an Adult-trained SegFormer and evaluate on Adult, Pediatric, and SSA data. Steering population-specific latents strongly affects in-domain performance but has weaker out-of-domain effects (Figure 5), indicating population-aligned causal bottlenecks. In contrast, steering universally shared latents yields minimal changes across datasets, comparable to random steering. Inspection via automated interpretation reveals that these universally shared latents predominantly encode stable anatomical, background, and boundary features, explaining their limited impact on tumor segmentation. Together, these results indicate that dataset shift arises from population-specific latent circuits rather than mismatched shared representations.

### 5.5. Baselines Comparison

We compare our cross-dataset latent diffing method against three baselines, Greedy matching, Sinkhorn alignment, and

*Table 2.* Unsupervised comparison of latent alignment methods between adults and pediatrics datasets.

| Method | Agreement ↑ | Stability ↑ | Collision ↓ | Predictability ↑ |
|--------|-------------|-------------|-------------|------------------|
| Greedy | 98.0 | 98.5 | 0.014 | 78.7 |
| Sinkhorn | 98.9 | 99.4 | 0.006 | 78.6 |
| Enc-only | 99.0 | 98.9 | 0.000 | 78.5 |
| **Ours** | **100.0** | **100.0** | **0.000** | **90.0** |

encoder-only Hungarian matching using four unsupervised metrics: *Agreement*, the fraction of latents whose matches agree between encoder- and decoder-based alignment; *Stability*, robustness under bootstrap resampling of the SAE dictionaries; *Collision*, the rate of many-to-one assignments; and *Predictability*, a functional specificity score measuring the gap between aligned similarity and a shuffled baseline (see subsection E.1). We do not use CrossCoder-based alignment due to known limitations: CrossCoders learn a shared projection that can entangle shared and model-specific features and exhibit instability across runs, which complicates correspondence interpretation (Mishra-Sharma et al., 2025; Dumas et al., 2025).

As shown in Table 2, Agreement and Stability are high for all methods, indicating that cross-dataset SAE alignment is well-conditioned. While Greedy, Sinkhorn, and encoder-only Hungarian recover largely consistent matches, they rely on a single projection space. Our method enforces cross-space consistency, yielding perfect agreement and substantially higher Predictability, indicating more specific and non-accidental correspondence.

## 6. Failure Diagnosis and Latent Intervention

We analyze whether segmentation failures can be diagnosed and corrected through representation-level analysis, focusing on instance-level and class-level weaknesses in SegFormer. We contextualize our approach using widely adopted inference-level refinements in BraTS pipelines, including connected-component (CCF) post-processing (Menze et al., 2014; Rai et al., 2024) and entropy-based uncertainty filtering (UF) from the BraTS uncertainty chal-

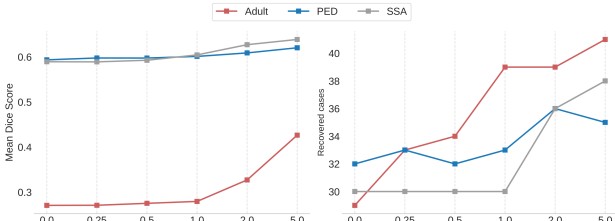

*Figure 6.* Effect of steering the most activated latent on failure cases across Pediatric (PED), Sub-Saharan African (SSA), and Adult glioma datasets. Amplifying these latents with increasing scaling strength improves the mean Dice score and increases the number of recovered cases.

*Table 3.* Mean Dice (%) on failure cases across datasets. We report absolute performance and improvement over the base model ($\Delta$).

| Intervention | Adult | PED | SSA |
|---|---|---|---|
| Base Model | 25.3 | 58.8 | 59.2 |
| CCF | 25.7 (+0.4) | 47.4 (−11.4) | 55.7 (−3.5) |
| UF | 27.5 (+2.2) | 47.5 (−11.3) | 56.4 (−2.8) |
| **Ours** | **43.0** (+17.7) | **62.0** (+3.2) | **63.8** (+4.6) |

lenge (Mehta et al., 2022; Mehrtash et al., 2020). These methods suppress spurious or low-confidence predictions and provide complementary points of comparison to our internal, causal interventions.

## 6.1. Instance-Level Failure Diagnosis

For each BraTS Adult, Pediatric, and Sub-Saharan African (SSA) dataset, we analyze the 60 lowest-performing cases as failure instances, selected to balance stability and specificity. For each case, we examine the top-$K$ most active sparse autoencoder (SAE) latents and observe systematic deviations in latent usage, with some latents abnormally amplified or suppressed.

**Latent Steering for Failure Recovery.** To test whether failures arise from latent miscalibration, we intervene on the most active SAE latents via latent steering. Given a latent vector $\mathbf{z}$ and selected indices $\mathcal{K}$, activations are scaled as

$$\tilde{z}_i = \begin{cases} \alpha z_i, & i \in \mathcal{K}, \\ z_i, & \text{otherwise,} \end{cases}$$

with $\alpha \in \{0.25, 0.5, 1.0, 2.0, 5.0\}$. As shown in Figure 6, increasing $\alpha$ recovers performance in 70.6%, 60.0%, and 63.3% of failure cases for Adult, Pediatric, and SSA datasets, respectively, substantially improving mean Dice without retraining.

**Comparison with Baselines.** Connected-component and entropy-based uncertainty filtering provide only modest

gains, improving some Adult cases but failing to recover most Pediatric and SSA failures Table 3. This reflects their nature, which suppresses false positives but does not restore missing structure.

**Interpreting Corrective Latents.** Automated interpretation shows that corrective latents predominantly encode background-related features; amplifying them suppresses spurious tumor responses. Such representation-level corrections are inaccessible to output-level baselines.

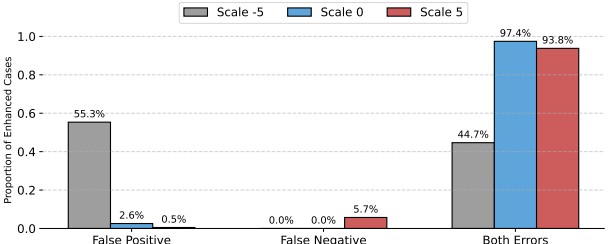

*Figure 7.* Failure-mode–specific effects of edema latent steering. Suppression primarily corrects false positives, amplification recovers false negatives, and both improve mixed-error cases, demonstrating complementary causal interventions.

## 6.2. Class-Level Performance Disparities

Beyond instance-level failures, the Adult model exhibits a systematic class-specific weakness: edema achieves a mean Dice of 65%, compared to 80% for enhancing tumor and 81% for necrotic core. We therefore focus on Adult cases with edema Dice below 50% (2,844 cases).

**Class-Specific Latent Intervention.** Using model diffing and automated interpretation, we identify edema-related latents and apply targeted steering via amplification, suppression, and zeroing (Figure 7). Error analysis reveals three dominant modes: false positives, false negatives, and mixed errors with suppression correcting hallucinations, amplification recovering missing edema, and zeroing best handling mixed cases.

*Table 4.* Edema and mean Dice (%) on Adult failure cases. Absolute scores and improvement over the base model ($\Delta$) are reported.

| Intervention | Edema Dice | Mean Dice |
|---|---|---|
| Base Model | 65.8 | 81.8 |
| CCF | 65.6 (-0.2) | 81.7 (-0.09) |
| UF | 65.8 (+0.0) | 81.8 (+0.0) |
| **Ours** | **69.1** (+3.2) | **82.6** (+0.79) |

Across the 2,844 failures, latent intervention recovers 1,344 cases (47.25%), improving edema Dice from 65% to 69% and overall mean Dice from 81.8% to 82.6 without retraining. Baselines provide limited benefit on this subset (Ta-

*Table 5.* Cross-domain segmentation performance between Adult and Pediatric cohorts, reporting per-class Dice (Classes 1–3), mean Dice, and edema precision (PR) and recall (RR), with absolute scores and improvement over baseline ($\Delta$).

| Method | Adult → Pediatrics | | | | | Pediatrics → Adult | | | | |
|---|---|---|---|---|---|---|---|---|---|---|
| | Necrotic | Edema | Enhancing | Mean Dice | PR | Necrotic | Edema | Enhancing | Mean Dice | RR |
| Base Model | **59.6** | 39.4 | 72.5 | 67.8 | 22.2 | 59.5 | 45.2 | 78.3 | 70.6 | 35.3 |
| CCF | 48.6 (-11) | 60.0 (+20) | 69.1 (-3) | 69.3 (+1) | 26.2 (+4) | 52.9 (-6) | 22.0 (-23) | 45.3 (-33) | 54.7 (-15) | 0.02 (-35) |
| UF | 59.6 (+0) | 39.6 (+.2) | 73.2 (+.7) | 68.0 (+.2) | 22.4 (+.2) | 59.7 (+.2) | 45.1 (-.1) | 78.3 (+0) | 70.6 (+0) | 35.1 (-.2) |
| **ALS (Ours)** | 57.5 (-2) | **74.2 (+34)** | **76.1 (+3)** | **76.9 (+9)** | **73.7 (+51)** | 55.0 (-4) | **49.3 (+4)** | 73.3 (-5) | 69.2 (-1) | **48.5 (+13)** |
| **MLS (Ours)** | 58.8 (-.8) | 52.3 (+12) | 71.8 (-.7) | 70.6 (+2) | 29.1 (+6) | **59.8 (+.3)** | 48.5 (+3) | **77.8 (-.5)** | **71.4 (+.8)** | 42.3 (+7) |

ble 4), as many failures are false-negative-dominated and require restoring missing internal features rather than suppressing uncertain predictions.

# 7. Out-of-Distribution Adaptation Without Retraining

Using model diffing, we separate domain-invariant SAE latents from population-specific latents, which act as causal bottlenecks under distribution shift. We study a realistic deployment setting: adapting a single trained segmentation model to out-of-distribution data *without retraining or access to target-domain models*. We evaluate two latent intervention strategies (Rimsky et al., 2024; Turner et al., 2023).

**Additive Latent Steering (ALS).** We inject a constant offset into selected SAE latents,

$$z_k \leftarrow z_k + \alpha, \quad k \in \mathcal{K}.$$

**Multiplicative Latent Steering (MLS).** We rescale selected latents,

$$z_k \leftarrow \alpha \, z_k, \quad k \in \mathcal{K},$$

modulating feature strength without introducing new signals.

We sweep $\alpha \in [-100, 100]$ for analysis purposes to characterize the effect of latent interventions across a wide range of steering strengths. We evaluate Adult→Pediatric and Pediatric→Adult adaptation, where performance degradation is consistently dominated by the edema class (Table 5). Using automated latent interpretation, we identify edema-controlling SAE latents and intervene on them at inference time.

The optimal strategy depends on adaptation direction. ALS is most effective for Adult→Pediatric, while MLS performs best for Pediatric→Adult (Figure 8). Adult→Pediatric errors are false-positive–dominated; suppressing edema latents improves Class 2 Dice from 39% to ∼74%. Pediatric→Adult errors are false-negative–dominated; amplifying the same latents restores missing predictions, improving Dice, mean Dice, and recall.

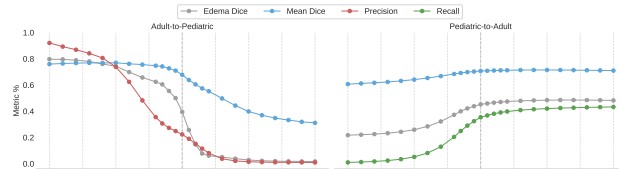

*Figure 8.* Domain adaptation under OOD shift. Negative scaling improves Adult→Pediatric performance, while positive scaling improves Pediatric→Adult performance, particularly for edema.

As shown in Table 5, Connected-component filtering partially mitigates false positives in Adult→Pediatric adaptation (60%), but underperforms latent steering (74.2%). Uncertainty filtering yields minimal gains. In Pediatric→Adult, neither baseline recovers false negatives, whereas latent amplification succeeds.

U-Net results are reported in Appendix G, confirming that the latent-intervention framework applies beyond Seg-Former.

# 8. Discussion

Our results show that SAE latents provide a useful mechanistic interface for diagnosing and correcting segmentation failures under dataset shift. However, several practical considerations remain. First, the intervention strength $\alpha$ is swept in our experiments only for analysis, to characterize controllability and causal effects of the identified latents. In real deployment, ground-truth Dice would not be available for tuning. Since the relevant latents are identified offline using training data and reused at inference time, $\alpha$ can instead be selected using label-free signals from the model output, such as predictive uncertainty, entropy of the predicted mask, consistency under test-time augmentations, or simple output statistics such as predicted lesion volume. The observed robustness across a range of $\alpha$ values further suggests that precise tuning may not always be necessary.

Second, our analysis uses a single mid-level layer for SAE extraction. This choice reflects a trade-off between spatial localization and semantic abstraction: early layers are often diffuse and background-dominated, while late layers are more task-specific and concentrated around tumor regions. Mid-level representations capture diverse and interpretable

failure modes, and the consistency of dataset-specific latents and intervention effects suggests that the observed mechanisms are not merely layer artifacts. Nevertheless, segmentation models operate across multiple scales, and extending our framework to multi-scale SAE alignment may reveal additional failure modes related to fine boundaries or global anatomical context.

Finally, while some of our interpretation metrics are described using brain-specific terminology, the underlying definitions are mask-based and anatomically agnostic. For example, "Brain Edge Preference" corresponds more generally to structure-boundary activation, and "Brain Depth" measures activation-weighted distance from a structure boundary. Our CT-ORG experiments demonstrate that the same pipeline can be applied to multi-organ CT without algorithmic changes.

## 9. Conclusion

We introduced a latent-level model diffing framework for medical image segmentation that enables direct comparison of internal representations across populations, revealing a clear separation between shared, population-invariant features and dataset-specific latent circuits that act as causal bottlenecks under distribution shift. By decomposing model activations with sparse autoencoders and aligning latents across independently trained models, we showed that systematic failures and cross-dataset performance gaps arise primarily from differential reliance on population-specific latents rather than a lack of shared representations, and that targeted latent-level interventions can recover failures, correct class-level disparities, and adapt models to out-of-distribution data without retraining or access to target-domain models. However, the latent features learned by SAEs are inherently sensitive to the training data distribution; although we analyze multiple cohorts spanning healthy anatomy and diverse tumor populations, different datasets may induce distinct or additional latent feature sets. Moreover, latent interpretation is constrained by the spatial and geometric metrics employed and and the availability of ground-truth masks, meaning that latents with complex, diffuse, or overlapping activation patterns may be only partially or imperfectly interpreted. Future work will explore richer alignment strategies that support graded or many-to-many correspondences, improved attribution of dataset-specific latents, and more expressive interpretation mechanisms that reduce reliance on handcrafted metrics and annotations.

## Acknowledgements

This research was enabled in part by support provided by Compute Ontario and the Digital Research Alliance of Canada.

## Impact Statement

This work introduces a mechanistic framework for analyzing and intervening on medical image segmentation models through latent-level model diffing. By decomposing internal representations into interpretable, population-aligned features, our approach enables systematic diagnosis of representation shift, identification of population-specific failure modes, and targeted correction of errors without retraining. The primary positive impact of this work lies in improving the reliability and equity of segmentation models under dataset and population shift. Our results show that performance degradation in underrepresented or out-of-distribution cohorts can often be traced to specific latent circuits and mitigated through causal interventions at inference time. This has the potential to reduce disparities in model performance across populations, particularly in settings where collecting large, representative datasets or retraining models is infeasible. Beyond medical imaging, the proposed framework contributes a general methodology for mechanistic model comparison and failure analysis, which may be applicable to other high-dimensional perception models. At the same time, we emphasize that these techniques are intended to support model auditing, robustness analysis, and scientific understanding, rather than to replace clinical judgment. Any deployment in high-stakes clinical environments requires rigorous validation, regulatory approval, and expert oversight.

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

## A. Dataset Statistics

We summarizes the key statistics of the datasets used in our framework, including domain characteristics, dataset size, and label composition Table 6. We consider both healthy and diseased cohorts. In healthy datasets, segmentation labels correspond to anatomical brain structures such as white matter, gray matter, and cerebrospinal fluid. In contrast, diseased datasets focus on pathological regions, where segmentation targets tumor subregions (e.g., necrotic core, edema, and enhancing tumor).

*Table 6.* Summary of datasets used in this study

| Dataset | Domain | # Volumes | # Classes |
| --- | --- | --- | --- |
| IXI | Healthy brain | 576 | 9 (8 subcortical + background) |
| BraTS Adult | Adult glioma | 1,251 | 4 (3 tumor + background) |
| BraTS Pediatric | Pediatric glioma | 99 | 4 (3 tumor + background) |
| BraTS SSA | SSA glioma | 60 | 4 (3 tumor + background) |

## B. Ablation Studies and Additional Analysis

We present additional ablation studies analyzing the sensitivity of our framework to key design choices. Specifically, we examine the effect of the encoder layer from which activations are extracted and the impact of the SAE expansion factor (dictionary size).

### B.1. Effect of Encoder Layer Selection

In the main paper, activations are extracted from a mid-level encoder layer. Here, we assess changes in representation by comparing early-, mid-, and late-stage encoder layers of the SegFormer model trained on the Pediatric cohort.

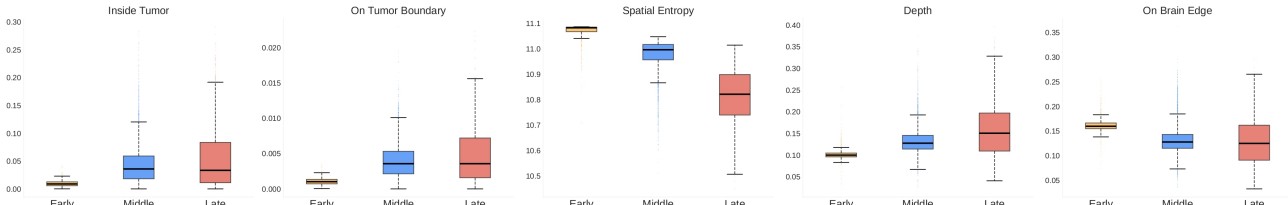

*Figure 9.* Layer-wise evolution of spatial properties in segmentation representations. SAE latent activations from early, middle, and late layers are compared using geometry-aware metrics.

As shown in Figure 9, geometry-aware metrics exhibit systematic variation across encoder depth. Early-layer latents show low inside-tumor and boundary ratios, together with high spatial entropy and strong alignment with the brain edge, indicating diffuse, globally distributed features that primarily encode coarse anatomical context. In contrast, mid-level representations exhibit increased sensitivity to tumor interiors and boundaries, reduced spatial entropy, and greater activation at intermediate brain depths, consistent with the emergence of localized, task-relevant features. Late-layer latents display the strongest inside-tumor activation and deepest brain localization, accompanied by the lowest entropy, reflecting highly concentrated representations associated with specific anatomical or pathological structures. For consistency and interpretability, we therefore adopt a mid-level encoder block in the main analysis.

Figure 10 provides qualitative examples of SAE latent activations across early, middle, and late encoder layers. Early-layer latents exhibit diffuse, globally distributed activations reflecting coarse anatomical context. Mid-level latents show more structured and localized responses, activating on tumor regions while also capturing surrounding anatomical context, with some features attenuating within tumors and emphasizing non-tumor regions. Late-layer latents are highly localized and strongly aligned with tumor regions, capturing concentrated, task-specific representations. Together, these examples qualitatively mirror the layer-wise trends observed in the geometry-aware metrics.

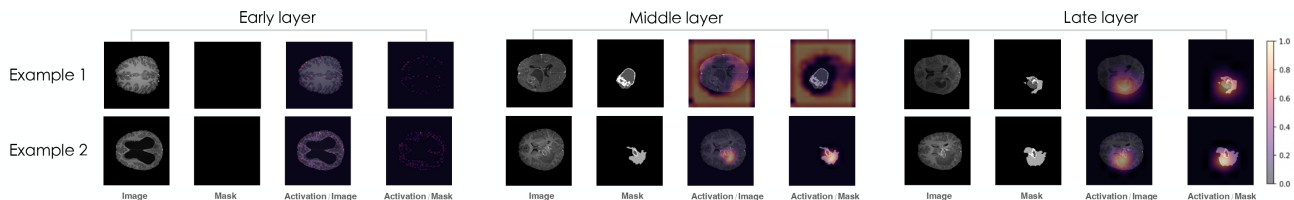

*Figure 10.* Qualitative comparison of SAE latent activations across encoder layers. As depth increases, latent activations transition from global anatomical patterns to increasingly localized and tumor-specific responses.

## B.2. Effect of Dictionary Size

To study the effect of representation capacity, we train SAEs with expansion factors of 8, 16, and 32, corresponding to progressively larger dictionary sizes. Figure 11 summarizes reconstruction quality, explained variance, and sparsity statistics for each setting on the adult dataset. Smaller expansion factors produce broader and less interpretable features, reflecting limited representational capacity and partial feature entanglement. In contrast, larger expansions encourage feature splitting, but at the cost of a substantially higher fraction of dead features, indicating over-parameterization.

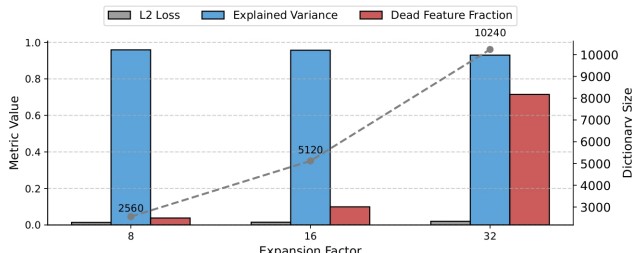

*Figure 11.* Effect of SAE dictionary size on reconstruction quality and feature sparsity. We evaluate sparse autoencoders trained with expansion factors of 8, 16, and 32, reporting reconstruction error (L2 loss), explained variance, and the fraction of dead features (bars), alongside the corresponding dictionary size (line, right axis). Smaller dictionaries underfit and produce broader, less specific features, while very large dictionaries lead to excessive feature fragmentation and high dead-feature rates. An intermediate expansion factor of 16 provides the best trade-off between reconstruction fidelity and interpretable, active latent features.

Across datasets, an expansion factor of 16 achieves the best balance between reconstruction fidelity, active feature utilization, and semantic interpretability. We therefore adopt this setting in all experiments.

## C. Auto-Interpretation Framework: Geometry- and Spatial-Based Metrics

**Automated Latent Characterization.** For each SAE latent, we analyze its Top-$K$ highest-activating samples and compute the following geometry- and anatomy-aware metrics:

- **Class Association.** Fraction of activation mass assigned to each segmentation class, $p_c = \sum h_{ij}\mathbb{I}[(i,j) \in M_c]/(\sum h_{ij} + \varepsilon)$.

- **Inside and Boundary Sensitivity.** Fraction of activation mass inside labeled regions and along their boundaries, $r_{\text{inside}}, r_{\text{boundary}} = \sum h_{ij}\mathbb{I}[(i,j) \in M, \partial M]/(\sum h_{ij} + \varepsilon)$, where $M$ denotes the anatomical segmentation.

- **Spatial Concentration.** Spatial entropy of the activation map, $\mathcal{H}(h) = -\sum \tilde{h}_{ij} \log(\tilde{h}_{ij} + \varepsilon)$, where lower values indicate localized activations.

- **Brain Edge Preference.** Fraction of activation mass near the brain boundary relative to total mass inside the brain, $r_{\text{edge}} = \sum h_{ij}\mathbb{I}[(i,j) \in E]/\sum h_{ij}\mathbb{I}[(i,j) \in B]$.

- **Brain Depth.** Mean activation-weighted depth from the brain boundary, with values near 0 indicating superficial and values near 1 indicating deep-brain features.

- **Coarse Spatial Localization.** Robust activation centroid estimated from high-activation pixels and expressed relative to the brain extent (left/right, top/bottom, quadrant).

- **Edge Angular Coverage.** Fraction of angular bins around the brain centroid containing high-activation pixels near the brain edge, capturing ring-like versus localized edge responses.



*Figure 12.* Visualization for some intermediate steps used in auto-interpretation. The top row shows the MRI slice, the extracted brain mask, the cortical edge band used for edge-based measurements. The bottom row illustrates the ground-truth tumor mask, the corresponding latent activation heatmap overlaid on the image, the estimated activation centroid. These spatial and anatomical signals are used to compute geometry-based metrics such as brain edge ratio, brain depth, spatial entropy, and centroid-based localization.

All metrics are computed per image and averaged across Top-$K$ samples, yielding a stable, interpretable descriptor for each latent used in automated semantic labeling and model diffing. Some visualizations of the auto-interp steps are presented in Figure 12.

## D. Qualitative Examples of SAE Latents

We present additional examples of SAE latents and their spatial activation patterns. As shown in Figure 13, multiple latents emerge for each tumor or anatomical category, capturing systematic variation in spatial location (e.g., left vs. right), morphology (e.g., diffuse vs. localized), scale, and depth within the brain. These examples demonstrate that SAEs decompose segmentation representations into fine-grained, clinically meaningful features that encode structure beyond mere class presence.

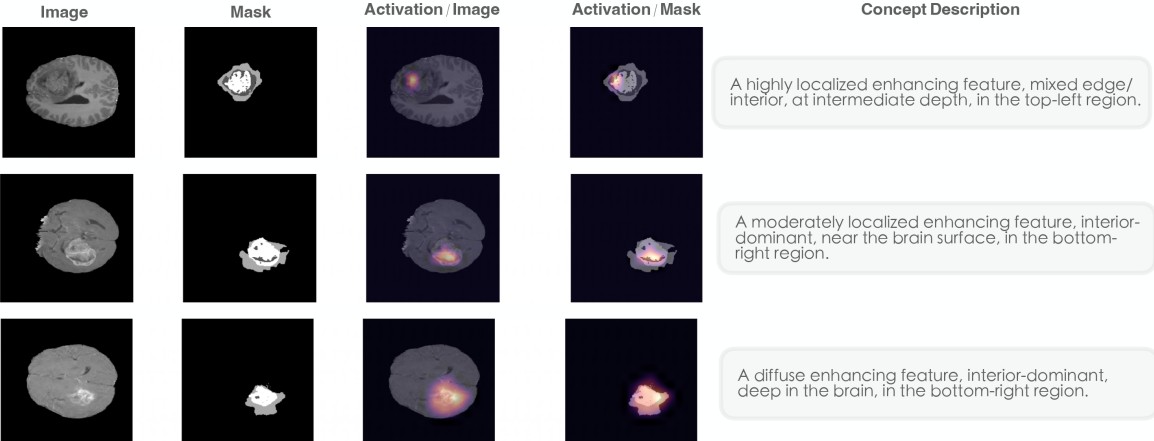

*Figure 13.* Different SAE latents linked to the same tumor class in the pediatric glioma model, capturing diverse spatial and morphological patterns.

## E. Identifying Shared Latents via Model Diffing

Table 7 reports the fraction of shared latents for each dataset pair. All pairs exhibit a non-zero overlap, indicating that models trained on different cohorts learn a partially shared internal basis. The largest overlap is observed between Adult and Pediatric models (58.9%), reflecting their closer anatomical and pathological similarity. Notably, the Adult–Healthy pair

also exhibits a relatively high overlap (36.1%), consistent with both datasets comprising adult brains. In contrast, the overlap is substantially lower for pairs involving the Sub-Saharan African cohort, despite also consisting of adult subjects. This suggests that factors beyond age, such as differences in acquisition protocols, scanners, or imaging conditions, may play a significant role in shaping learned representations, consistent with prior work on MRI acquisition and domain variability (Fortin et al., 2017; Shinohara et al., 2014).

*Table 7.* Fraction of shared SAE latents between dataset pairs.

|  | **Adult** | **Pediatrics** | **Sub-Saharan** | **Healthy** |
|---|---|---|---|---|
| **Adult** | – | 58.9% | 43.2% | 36.1% |
| **Pediatrics** | 58.9% | – | 33.9% | 28.9% |
| **Sub-Saharan** | 43.2% | 33.9% | – | 19.2% |
| **Healthy** | 36.1% | 28.9% | 19.2% | – |

**Feature Swapping.** To probe the functional role of shared latents, we perform feature swapping from the Healthy (IXI) model into the Adult glioma model. Specifically, SAE latents identified as shared are replaced with their aligned counterparts from the Healthy model, and their behavior is compared to swaps involving non-shared latents. As shown in Figure 14, swapping shared latents preserves spatial activation structure and anatomical alignment in the Adult model, whereas swapping non-shared latents leads to disrupted or absent activations. These results provide causal evidence that shared latents correspond to transferable internal mechanisms, while non-shared latents capture dataset-specific representations.

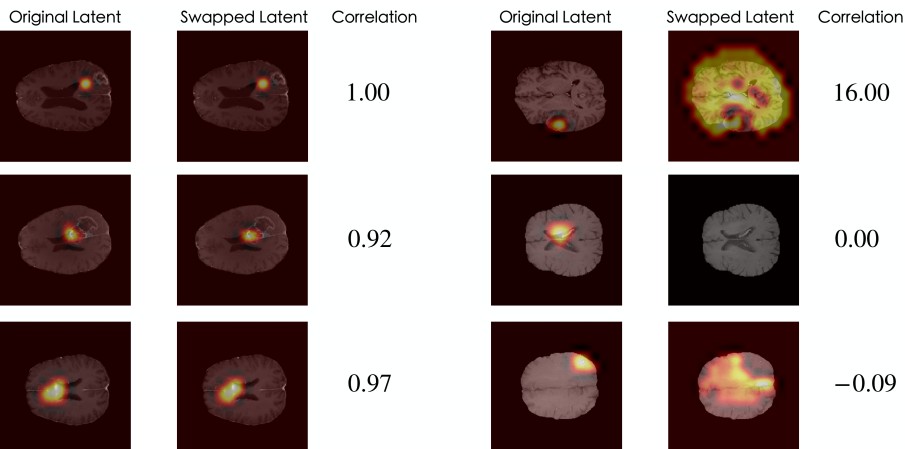

*Figure 14.* Feature swapping between Adult and Healthy SAEs. Shared latents (left) preserve activation structure and spatial correlation, while non-shared latents (right) produce disrupted or absent activations.

### E.1. Details of Alignment Evaluation Metrics

We evaluate latent alignment quality using four unsupervised metrics designed to probe complementary aspects of correspondence without relying on labels or downstream task performance.

**Agreement.** Agreement measures the fraction of latents for which the encoder-based and decoder-based alignment procedures identify the same matched counterpart. This metric captures internal consistency across representation spaces and tests whether a correspondence is stable under different linear projections of the SAE dictionary.

**Stability.** Stability evaluates robustness of the alignment under perturbations by repeatedly recomputing matches on bootstrap subsamples of the SAE dictionaries. A latent is considered stable if it is matched to the same counterpart across resampled trials. High stability indicates that the correspondence is not sensitive to small changes in the feature set and reflects a well-conditioned alignment problem.

**Collision.** Collision measures the rate of many-to-one assignments, where multiple source latents are matched to the same target latent. This metric quantifies ambiguity in the alignment and distinguishes strict bijective matchings from greedy or

soft assignments that allow overlap.

**Predictability.** Predictability assesses the functional specificity of the alignment by comparing the mean similarity of matched latent pairs against a shuffled baseline. Formally, it measures the gap between aligned similarity and the expected similarity under random pairing. Higher values indicate that the identified correspondences are non-accidental and reflect meaningful shared structure rather than coincidental similarity.

These metrics provide a label-free evaluation of alignment quality, distinguishing surface-level similarity from robust, mechanistically consistent correspondence. We compare alignment quality across Adult–SSA and Adult–IXI dataset pairs, and observe consistent trends across settings, as reported in Table 8.

*Table 8.* Unsupervised latent alignment comparison across dataset pairs. Agreement and Stability are reported as percentages.

| Dataset Pair | Method | Agreement ↑ | Stability ↑ | Collision ↓ | Predictability ↑ |
|---|---|---|---|---|---|
| Adult–SSA | Greedy | 97.6 | 98.9 | 0.019 | 72.78 |
| | Sinkhorn | 98.5 | 99.3 | 0.010 | 72.72 |
| | Enc-only | 98.8 | 98.6 | 0.00 | 72.6 |
| | **Ours** | **100.0** | **100.0** | **0.000** | **88.5** |
| Adult–IXI | Greedy | 98.0 | 98.1 | 0.018 | 70.3 |
| | Sinkhorn | 98.7 | 98.9 | 0.009 | 70.2 |
| | Enc-only | 99.2 | 99.0 | 0.00 | 70.1 |
| | **Ours** | **100.0** | **100.0** | **0.00** | **90.1** |

# F. Beyond Brain MRI

To assess whether our interpretation metrics generalize beyond brain MRI, we first evaluate the same pipeline on CT-ORG, a multi-organ CT dataset (Rister et al., 2020). This setting differs in both imaging modality and anatomical targets. Importantly, our automated interpretation metrics require only segmentation masks and therefore remain anatomically agnostic. For example, *Brain Edge Preference* directly becomes *Organ Edge Preference*, measuring activation near the structure boundary versus the interior, while *Brain Depth* becomes *Depth Within Structure*. Other metrics, such as class association and spatial concentration, are already fully mask-based. We apply the pipeline to CT-ORG without algorithmic changes Figure 15, only renaming the metrics to reflect the target anatomy. This demonstrates that the proposed latent interpretation framework can transfer across organs and modalities with minimal adaptation.

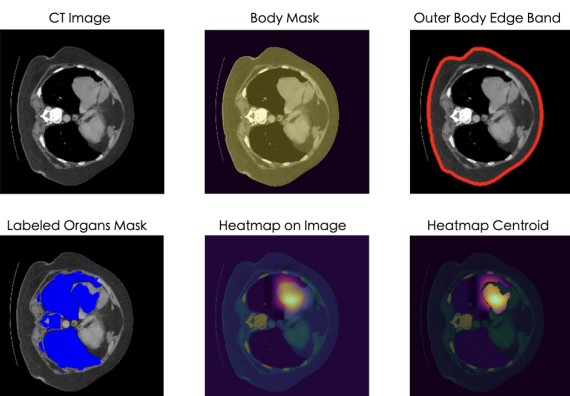

*Figure 15.* Intermediate auto-interpretation steps on CT-ORG. We show the CT slice, organ mask, edge band, ground-truth mask, latent activation overlay, and activation centroid used to compute geometry-based metrics.

As an additional sanity check outside medical imaging, we evaluate the method on CelebAMask-HQ (Lee et al., 2020), a face parsing dataset with 30,000 high-resolution images and 19 facial-part and accessory classes, including skin, eyes, nose, mouth, hair, glasses, earrings, and clothing. Compared to medical datasets, CelebAMask-HQ contains different visual statistics, more appearance variability, and frequent accessory-induced occlusions.

**Training.** For each dataset, we train a SegFormer model using Dice-based segmentation objectives. We then extract activations from a fixed intermediate encoder layer and train a BatchTopK SAE with sparsity $k = 32$ and expansion factor 16.

**Latent Interpretation.** Applying the same automated interpretation pipeline, we find that SAE latents align with meaningful structures in both CT-ORG and CelebAMask-HQ Figure 16. In CT-ORG, latents correspond to organ-specific concepts across multiple anatomical structures. In CelebAMask-HQ, latents align with facial components and accessories, and also show intra-class disentanglement, such as separate eye-related latents for subjects with and without glasses.

**Summary.** These experiments show that our latent analysis and auto-interpretation pipeline is not limited to brain MRI. It can be applied across different organs, imaging modalities, and even non-medical semantic segmentation settings, while preserving meaningful spatial and semantic structure.

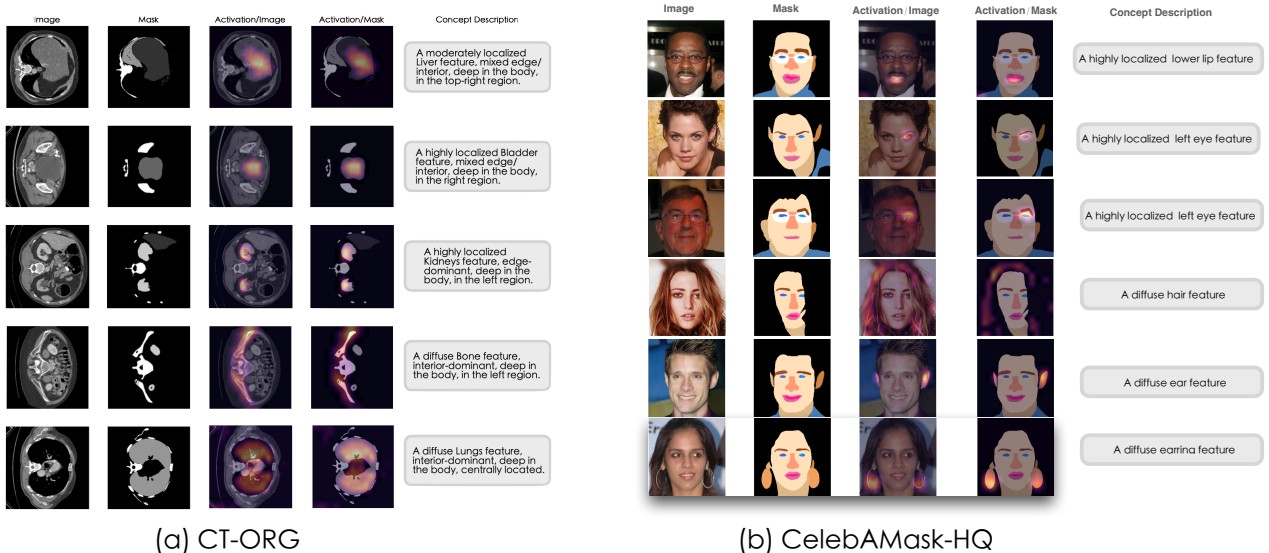

(a) CT-ORG  (b) CelebAMask-HQ

*Figure 16.* Qualitative examples of automated latent interpretation on CT-ORG and CelebAMask-HQ. The learned latents capture meaningful spatial structures across different organs, modalities, and non-medical semantic classes.

## G. Out-of-Distribution Adaptation U-Net

Figure 17 reports latent-steering results for Adult↔Pediatric transfer using U-Net. Amplifying the relevant SAE latents improves performance in both adaptation directions, with consistent trends indicating that these latents act as causal bottlenecks under distribution shift.

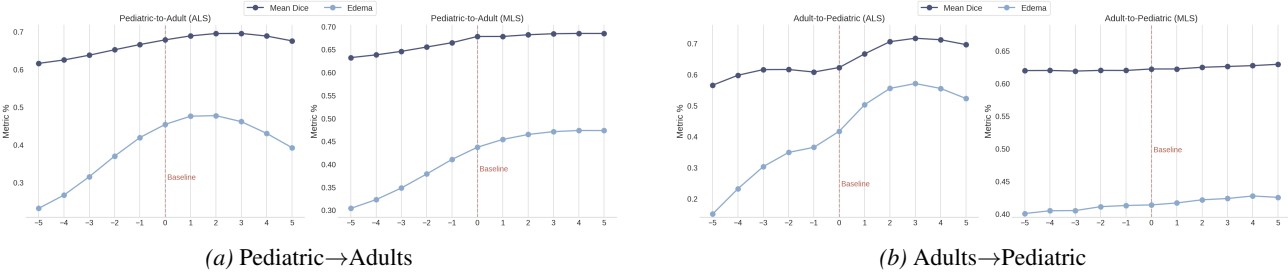

*(a)* Pediatric→Adults  *(b)* Adults→Pediatric

*Figure 17.* Latent amplification improves Adult↔Pediatric transfer, supporting causal bottlenecks.

