# OpenReview forum: "Med-SegLens: Latent-Level Model Diffing for Interpretable Medical Image Segmentation"
_ICML.cc/2026/Conference — ICML 2026 regular_

### Official Review · Reviewer_cxke · 2026-03-06

**Soundness:** 3
**Presentation:** 3
**Significance:** 3
**Originality:** 3
**Overall Recommendation:** 4
**Confidence:** 3

**Summary:**

The authors proposed an interpretability analysis method for medical image segmentation models, which leverages mechanistic interpretability to interpret model learning, and leverages the Hungarian algorithm to compare models on medical images with different populations (e.g., adult and pediatric glioma) at the latent level to study the distinctions the models learned from dataset distributions. Experiments are conducted using SegFormer and U-Net on brain MRI datasets. The results demonstrate that the proposed method can uncover how models represent anatomical structures, why failure arises under dataset shifts, and how targeted latent interventions can correct these failures.

**Compliance With Llm Reviewing Policy:**

Affirmed.

**Final Justification:**

The authors have addressed my concerns, therefore, I changed my evaluation.

**Key Questions For Authors:**

1. What is the rationale behind the design of the geometry- and spatial-based metrics?
2. What are the main problems to tackle if the framework is used in other medical image modalities?
3. In U-Net, a large amount (96.4%) of latents are background (Figure 3). The authors explain that this is due to non-tumor anatomy in MRI. Does this situation remain the same across all layers in the model encoders?
5. Why do the authors only leverage SegFormer for across-dataset experiments? Will the conclusions be suitable for U-Net as well?
5. In Figure 2, which image shows representative SAE latents from SegFormer?

**Limitations:**

yes

**Strengths And Weaknesses:**

**Strengths**
1. The framework is technically sound; the designed combination of a sparse autoencoder (to extract sparse latent representations) and the proposed automatic latent semantics discovery module (to map those SAE latents into semantics by reverting latent activations back to spatial resolution and calculating geometry- and spatial-based metrics) is naturally fluent and elegant.
2. The presentation is good. The figures are clear and easy to follow, the experiments and analysis are well designed and comprehensive. The manuscript is well-written.
3. The motivation is clear, and this problem has its significance (interpretability for medical image segmentation and correcting errors without retraining). Theoretically, the proposed method could be applied to other imaging modalities and datasets.
4. The work introduces a new combination of mechanistic interpretability (to uncover the latents) and model diffing (matching between different models trained on different distributions), and this combination is well-articulated.

**Weaknesses**

1. The motivation of the work is interpretability for medical image segmentation. However, the experiments are restricted to brain MRI datasets and glioma, although in the supplementary, the authors tested the framework on a natural image dataset CelebAMask-HQ, it does not suit the scope of the work "medical image" very well. Experiments on other kinds of medical images (e.g., other modalities, organs, diseases) will help validate the claim.
2. The automated interpretation pipeline relies on human-defined geometry- and spatial-based metrics to bridge clinically meaningful concepts and latents, which may not be easy to define in more complex medical image modalities.
3. The Hungarian Algorithm is designed for bipartite graph maximum matching, which holds an assumption that the matching is between one latent from each dataset. However, in different datasets, due to distribution shift, the one-to-one mapping may not be ideal, and the significance of each latent may be different.

---

> ### Author Rebuttal · Authors · 2026-03-30
>
> We thank Reviewer cxke for their thorough feedback. We address each point below.
>
> ---
> **1. Rationale for geometry/spatial Metrics**
>
> The geometry- and spatial-based metrics provide a quantitative way to interpret SAE latents in medical imaging, where semantics are tied to spatial location and anatomy. Unlike natural images, meaningful concepts (tumor vs. healthy tissue, boundary vs. interior, ...) are spatially grounded, so qualitative inspection alone is insufficient.
>
> Our metrics capture complementary aspects of latent behavior:
> * Anatomical alignment (Class Association, Inside/Boundary Sensitivity): links latents to clinically meaningful regions.
> * Geometric context (Edge Preference, Depth): distinguishes boundary vs. interior structures.
> * Spatial distribution (Spatial Concentration, Centroid Localization): characterizes whether activations are localized or diffuse.
> * Structural patterns (Edge Angular Coverage): captures higher-level spatial organization ring-like or directional responses.
>
> This design enables automated, dataset-aware semantic labeling of latents, supporting downstream tasks such as model diffing and targeted interventions. While alternatives (VLM/LLM-based descriptions) are possible, our metrics provide structured, grounded signals that can guide such models, improving consistency and reducing hallucination.
>
> ---
> **2. Challenges for other modalities**
>
> Thank you for raising the important question. While our main experiments focus on MRI to study dataset shift under controlled conditions (same anatomy, varying datasets), we additionally evaluate on a different modality using the CT-ORG dataset [1]. The pipeline remains unchanged across modalities: Kindly see Reviewer RvdD 'c' for CT-ORG results.
>
> The main challenges when extending to other modalities are not in the interpretation framework, but in: segmentation quality, modality-specific intensity distributions, and anatomical variability affecting metric expression (not definition).
>
> ---
> **3. U-Net latents Evaluation**
>
> We report results from a mid-level U-Net encoder layer, motivated by prior work [2, 3, 4] showing intermediate layers best balance semantic richness and spatial localization. We further analyze early and late layers [Fig3](https://tinyurl.com/5czc2uxw) and observe consistent, interpretable trends: early layers capture diffuse, low-level features, mid layers are more balanced but still background-dominated, and late layers become increasingly task-specific with stronger tumor representation.
>
> This follows the hierarchical nature of deep models: early layers encode general anatomy, while deeper layers specialize toward task-relevant regions. Thus, background dominance decreases with depth but remains prevalent. Our goal is not to compare U-Net vs. SegFormer, but to analyze their representational strategies; overall conclusions remain consistent.
>
> ---
> **4. U-Net for cross-dataset**
>
> Thank you for the question. We use SegFormer to isolate dataset shift effects within a single architecture, not to claim architectural superiority. To verify generality, we repeat experiments with U-Net and observe consistent results:
> - Latent alignment [Fig4](https://tinyurl.com/5f7wk79z): Shared latents preserve structure, while non-shared latents disrupt it, matching SegFormer.
> - Latent steering [Fig5](https://tinyurl.com/5ax7ze4s) / [Fig6](https://tinyurl.com/44aswua6): For Adult ↔ Pediatric, amplifying relevant latents improves performance in both directions, with consistent trends confirming causal bottlenecks.
>
> Overall, shared/dataset-specific structure and effective latent interventions adapts to U-Net.
>
> ---
> **5. Figure 2 SAE latents**
>
> Thank you for pointing this out. We agree Figure 2 does not clearly indicate which latents correspond to each model and will revise it to explicitly label the source architecture. In Fig. 2, columns 1, 3, 4, and 5 correspond to SegFormer, and the remaining columns to U-Net. Figures 10 and 13 also show SegFormer latents. We additionally include representative U-Net latents [Fig7](https://tinyurl.com/5kt8dxx8).
>
> ---
> **6. Hungarian Algorithm Matching**
>
> We use the Hungarian algorithm to enforce a strict, interpretable one-to-one mapping, avoiding many-to-one matches that introduce entanglement and hinder identification of shared vs. dataset-specific features. To account for imperfect alignment, we apply conservative filtering: agreement between encoder/decoder similarities and a threshold (τ = 0.8). Latents that fail these criteria are left unmatched and treated as dataset-specific. While more flexible methods (e.g., CrossCoders) could model richer relationships, they often introduce feature entanglement and instability, reducing interpretability. Our constrained matching provides a more stable and reliable basis for latent-level comparison.
>
> ---
> [1] Blaine, et al., s41597-020-00715-8
>
> [2] Skean et al., arXiv:2502.02013.
>
> [3] Minder et al., arXiv:2504.02922.
>
> [4] Wang et al., arXiv:2506.19823.

---

> > ### Author Rebuttal · Reviewer_cxke · 2026-04-02
> >
> > I appreciate the responses and they addressed my concerns.

---

### Official Review · Reviewer_HLUw · 2026-03-11

**Soundness:** 2
**Presentation:** 3
**Significance:** 2
**Originality:** 3
**Overall Recommendation:** 4
**Confidence:** 3

**Summary:**

This manuscript introduces Med-SegLens, a framework designed to decompose the intermediate activations of medical image segmentation models, specifically SegFormer and U-Net, into semantically interpretable latent features using Sparse Autoencoders (SAEs). Through a systematic model-diffing analysis across diverse cohorts, including healthy individuals and adult, pediatric, and sub-Saharan African glioma patients, the authors identify stable architectural representations and features driven by dataset shift. Furthermore, the framework enables causal interventions through latent-level ablation or steering, allowing for the correction of segmentation errors without requiring model retraining.

**Compliance With Llm Reviewing Policy:**

Affirmed.

**Key Questions For Authors:**

- Computational Overhead and Deployment Complexity

The training of SAEs and the associated latent alignment processes demand immense computational resources, frequently surpassing the costs required to train the primary segmentation models. In a clinical deployment setting, the necessity of maintaining an extensive "SAE forest" to interpret specific masks introduces significant concerns regarding the return on investment given the high computational expenditure.

- Ambiguity in the Clinical Translational Path

While the manuscript successfully demonstrates technical controllability, the practical application within a radiological workflow remains insufficiently addressed. Clinicians typically lack the technical background to interact with abstract latent features directly. The paper would be strengthened by a more thorough discussion on how this framework might be transformed into an intuitive and efficient tool for clinical decision support.

**Limitations:**

See Above.

**Strengths And Weaknesses:**

Strenght:

- Experimental Rigor and Diversity

The study is distinguished by its comprehensive experimental design, which encompasses a wide array of clinical cohorts and distinct model architectures. The detailed alignment of features across various populations provides a robust validation of the framework's capacity to detect and characterize data distribution shifts in medical imaging.

- Transition from Observation to Active Intervention

A significant contribution of this work is its progression beyond mere post-hoc explanation of model behavior. By demonstrating that the direct manipulation of latent features can enhance model performance, the authors establish a viable path for using latent steering to improve the robustness of medical AI systems.

---

> ### Author Rebuttal · Authors · 2026-03-30
>
> We thank Reviewer HLUw for their thorough feedback. We address each point below.
>
> ---
> **a. Computational Overhead and Deployment Complexity.**
>
> We would like to clarify that while SAEs are trained on activations of the segmentation model, this process is performed offline, after the segmentation model has been trained, and does not interfere with or slow down the standard training or inference pipeline. In other words, SAE training constitutes an analysis layer on top of a fixed model, rather than part of the core segmentation workflow.
>
> In terms of cost, SAE training is (i) more lightweight than training the base model, as it operates on intermediate activations, (ii) highly parallelizable, and (iii) a one-time step per model. It is not required during inference unless interpretability or latent intervention is used. Deployment also does not require a large “SAE forest” a single SAE per model (or domain) is sufficient and used only as a diagnostic/adaptation tool.
>
> At inference time, the overhead remains minimal. Latent interventions (e.g., scaling or shifting selected latents) are simple operations, adding only a small constant cost relative to standard model inference. We provide a quantitative comparison in [Table1](https://tinyurl.com/y27phj5n), showing that while there is a modest increase in runtime, the per-image latency remains low and practical
>
> Importantly, this one-time cost enables capabilities otherwise difficult to achieve: fine-grained interpretability, identification of dataset-specific failure modes, and training-free adaptation under distribution shift. These properties are particularly valuable in clinical settings, where transparency is essential and black-box models are often insufficient [1].
>
> Overall, SAEs provide a modular analysis and adaptation layer, combining minimal inference overhead with substantial gains in interpretability and controllability.
>
> ---
> **b. Ambiguity in the Clinical Translational Path.**
>
> Thank you for this important comment. We agree that translating latent-level analysis into a clinically usable workflow is critical and deserves further clarification.
>
> Our framework is not intended for clinicians to directly interact with raw latent features. Instead, Med-SegLens is designed as a backend interpretability and control layer that can be integrated into existing radiological systems through intuitive interfaces.
> We envision several clinically meaningful UI integrations:
> - **Confidence Heatmaps**: Latent activations associated with uncertainty or known failure modes can be visualized directly on the scan as overlays, highlighting regions where the model is less reliable (e.g., boundary ambiguity or missing structures).
> - **Automated Triage Alerts**: Dataset-specific failure latents can trigger automatic alerts such as “potential under-segmentation in edema region” or “low-confidence prediction”, enabling prioritization of cases for review without requiring clinicians to interpret model internals.
> - **Region-Level Explanations**: Instead of exposing latent IDs, we map them to clinically meaningful concepts (e.g., edema, tumor boundary, background leakage), which can be presented as structured annotations alongside the segmentation output.
>
> In addition, Med-SegLens supports:
> - **Automated Error Detection and Quality Control**: Failure-associated latents can be used to flag unreliable predictions and assist in triaging difficult cases.
> - **Lightweight, Model-Side Adaptation** (No User Interaction Required): Latent interventions (e.g., scaling or suppressing specific features) can be applied automatically at inference time based on predefined signals (e.g., uncertainty), improving predictions without requiring clinician input.
>
> From a system perspective, this positions our framework as:
> - A model introspection and correction layer for developers and clinical ML engineers, and
> - A decision-support enhancement layer for clinicians, through improved outputs, uncertainty cues, and intuitive visual explanations.
>
> We agree that building a fully user-facing clinical tool is a crucial next step (currently working on an open-source tool). Our current work focuses on establishing the mechanistic foundation and controllability, which we view as a prerequisite for safe and effective clinical deployment. We will revise the manuscript to better articulate this translational pathway and provide more concrete examples of how latent-level analysis can be surfaced in an intuitive, non-technical manner.
>
> ---
>
> [1] C Rudin et al., arXiv:1811.10154

---

### Official Review · Reviewer_RvdD · 2026-03-12

**Soundness:** 3
**Presentation:** 3
**Significance:** 3
**Originality:** 3
**Overall Recommendation:** 5
**Confidence:** 5

**Summary:**

This paper creatively introduces mechanistic interpretability and sparse autoencoder (SAE) technologies, which are popular in the field of large models, into the task of dense medical image segmentation, proposing the Med-SegLens framework. By extracting activations from the intermediate layers of segmentation networks (such as SegFormer and U-Net) and training SAEs, this framework decouples black-box representations into interpretable latent features. In addition, this method supports cross-architecture and cross-dataset feature alignment, which not only achieves the diagnosis of model failure modes but also proposes an out-of-distribution (OOD) adaptive latent steering strategy that does not require retraining.

**Compliance With Llm Reviewing Policy:**

Affirmed.

**Final Justification:**

I appreciate the authors' detailed and constructive responses to my concerns. The authors have demonstrated the practical viability and cross-domain robustness of their interpretable framework. They have fully addressed my concerns.

**Key Questions For Authors:**

When exploring retraining-free out-of-distribution (OOD) adaptation (Section 7), the paper proposes applying additive (ALS) or multiplicative (MLS) steering to selected SAE latent features, and scanning the scaling coefficient $\alpha$ within the range of [-100, 100] to search for the optimal performance. However, in strictly unsupervised OOD scenarios during real-world clinical deployment, the model cannot access the true segmentation labels (Ground Truth) of the target domain. This means that in practical applications, it is impossible to retroactively select the optimal intervention intensity $\alpha$ by calculating the Dice score. Without label feedback from the target domain, this intervention strategy will face significant deployment challenges. It is recommended to propose or evaluate a heuristic strategy that does not require target domain labels (such as predictive uncertainty or entropy minimization based on model outputs) to automatically determine and calibrate the appropriate intervention intensity $\alpha$, thereby demonstrating the practical viability of this method in real clinical environments.


The paper chooses to extract activations specifically from the "middle layer" of the segmentation model to train the Sparse Autoencoder (SAE). Although the spatial attribute evolution analysis in Appendix B.1 suggests that the middle layer possesses good localization and semantic properties, the predictions of modern medical segmentation networks (such as U-Net and SegFormer evaluated in this paper) rely heavily on the fusion of multi-scale features. High-resolution boundary details encoded by shallow networks or global pathological contexts encoded by deep networks may similarly experience degradation during dataset shifts. Performing latent alignment and Model Diffing solely on a single middle layer carries the risk of missing key failure modes driven by other network layers. It is recommended to further explore the joint alignment of multi-scale/cross-layer SAEs, or to deeply discuss the inherent limitations of such a single-layer extraction strategy in multi-scale dense prediction tasks.


The paper proposes an automated latent feature interpretation pipeline based on geometric and spatial metrics, utilizing heuristic indicators that heavily rely on brain anatomical priors, such as the "brain-edge ratio" and "depth within brain." While this method achieves impressive interpretation results on brain MRIs (IXI and BraTS datasets), these hard-coded geometric features—which are highly dependent on specific organ anatomy—may significantly restrict the generalization ability of the entire framework to other medical image segmentation tasks (e.g., multi-organ segmentation in abdominal CTs or lesions with high deformation). It is recommended that the authors provide an in-depth analysis of the generalization limitations of this pipeline in the discussion section, or further clarify how these heuristic rules could be transferred with minimal cost to non-brain imaging domains that possess entirely different spatial distribution characteristics.


In the cross-dataset SAE feature alignment, the paper applies the Hungarian algorithm based on a cosine similarity matrix and establishes a strict similarity threshold of $\tau=0.8$ to define shared latents. Meanwhile, Appendix Figure 11 shows that with an expansion factor of 16, a certain proportion of "Dead Features" still exists in the SAE dictionary. Two technical details need to be clarified here: First, prior to executing the Hungarian matching, were these unactivated dead features explicitly filtered out? If not, they might interfere with the optimal assignment during global matching. Second, the hard cutoff at $\tau=0.8$ might misclassify "slightly misaligned shared features" (caused by the inherent randomness of the model optimization process) as "dataset-specific latents." It is recommended to include a more comprehensive sensitivity analysis experiment on the similarity threshold $\tau$ to quantify the actual impact of this parameter on the proportion of shared features and the subsequent intervention effects.

**Limitations:**

The primary limitation of this method is that its automated latent feature interpretation pipeline is heavily bound to hard-coded anatomical priors specific to particular organs (such as the brain-edge ratio). Consequently, it is difficult to smoothly generalize this highly promising interpretable framework to non-brain medical image segmentation tasks characterized by high deformation or the lack of a fixed anatomical reference frame.

**Strengths And Weaknesses:**

Strengths
This work is highly pioneering in its perspective, successfully transferring SAE and model diffing from the NLP domain to medical image segmentation, providing a highly promising new research paradigm for understanding distribution shifts and shortcut learning in medical AI. More importantly, the framework does not stop at merely explaining why the model fails; it achieves a complete closed loop from explanation to intervention, where its proposed latent steering allows for the direct correction of segmentation results by fine-tuning specific latent feature coefficients during the inference stage, demonstrating extremely high practical clinical application value.

Weaknesses
The automated interpretation pipeline of this method relies heavily on hard-coded heuristic geometric priors such as brain anatomical structures, and this strong assumption may be difficult to smoothly transfer to segmentation tasks of abdominal multi-organs or highly deformable lesions, limiting its generalization ability. Additionally, modern dense prediction networks rely heavily on the fusion of multi-scale features, whereas this paper only performs SAE dictionary learning on a single intermediate layer, which may miss key failure modes caused by shallow-layer boundary degradation or deep-layer semantic collapse. In real-world clinical blind-testing OOD adaptive scenarios, since the model cannot access target domain ground truth labels, it is difficult to reversely guide the selection of optimal scaling coefficients, which also poses a challenge to the feasibility of its fully automated deployment.

---

> ### Author Rebuttal · Authors · 2026-03-30
>
> We thank Reviewer RvdD for their thorough feedback. We address each point below.
>
> ---
> **a. Intervention strength without labels.**
>
> Thank you for the insightful comment. We agree that selecting the optimal intervention strength (α) using ground-truth Dice is not feasible in a fully unsupervised deployment setting. In our experiments, the sweep over α ∈ [−100, 100] is used to characterize the effect of latent interventions and demonstrate controllability, rather than to define a deployment-time tuning procedure.
>
> Importantly, the latents themselves (e.g., edema-related features) are identified once offline using training data and then reused at inference time for intervention without requiring labels. The role of α is simply to modulate the strength of these pre-identified features.
>
> In practice, α can be selected using label-free heuristics based on model outputs, as suggested by the reviewer. For example,one can monitor predictive uncertainty (e.g., entropy of the output mask), consistency under augmentations, or simple output statistics such as predicted edema volume to detect over- or under-segmentation and adjust α accordingly. Additionally, we observe that performance improvements are often stable across a range of α values, suggesting that precise tuning is not required.
>
> We will clarify in the paper that the α sweep is intended for analysis, and discuss label-free strategies for selecting α in real deployment scenarios.
>
> ---
> **b. Single middle layer instead of multi-scale features.**
>
> Thank you for the insightful comment. We agree that segmentation models rely on multi-scale features (early: boundaries, late: global context). We use a single mid-level layer as a principled trade-off: Appendix B.1 shows early layers are diffuse and background-dominated, while late layers are highly task-specific and concentrated on tumor regions. Mid-level features balance semantic and spatial information, making them more suitable for capturing diverse failure modes.
>
> Moreover, representations evolve smoothly across depth, so nearby layers are expected to be similar [1]; combining them may introduce redundancy and feature entanglement, reducing interpretability [2]. This aligns with prior work showing intermediate layers capture rich, analyzable structure [3, 4, 5]. We acknowledge that single-layer analysis may miss some failure modes (e.g., fine boundaries or global context). However, key effects: dataset-specific latents and effective interventions, are already consistent at this level, indicating we capture meaningful representation differences rather than layer artifacts. Extending to multi-scale SAE alignment is a promising direction, which we will clarify as future work.
>
> ---
> **c. Do anatomy-specific heuristics limit generalization beyond brain MRI.**
>
> Thank you for this important observation. We agree that terms like Brain Edge & Brain Depth may suggest limited generalizability, but the metrics are anatomically agnostic by design and rely only on a segmentation mask.
>
> For example, Brain Edge Preference measures activation near the structure boundary vs. inside (i.e., a general Structure Edge Preference), and Brain Depth captures activation-weighted distance from the boundary (Depth Within Structure). Other metrics (e.g., class association, spatial concentration) are already fully mask-based and organ-agnostic.
>
> We validate this by applying the same pipeline to CT-ORG (multi-organ CT) [6] without any algorithmic changes, only renaming metrics (e.g., Brain Edge→Organ Edge). See pipeline illustration [Fig1](https://tinyurl.com/2wb6e36p) and example latent concepts [Fig2](https://tinyurl.com/5ap8kmrb). This demonstrates minimal generalization cost.
>
> We acknowledge that highly irregular or deformable structures (e.g., diffuse lesions) may reduce stability of some metrics (e.g., edge coverage), and will clarify these limitations in the discussion.
>
> ---
> **d. Robustness of alignment to dead features & similarity threshold**
>
> Thank you for these important questions.
>
> - Dead features. Inactive latents are filtered out prior to alignment (based on negligible activation), so Hungarian matching operates only on active features and is not affected by dead latents.
>
> - Similarity threshold (τ). We use a conservative criterion: latents must have high similarity in both encoder and decoder spaces, with τ = 0.8 chosen via a sweep to balance stability and coverage. Lower τ increases matches but adds noise; higher τ yields fewer, more reliable pairs. Importantly, key findings (dataset-specific latents and intervention effects) remain consistent across this range.
>
> We agree that making this more explicit would strengthen the paper, and we will include a brief sensitivity analysis of τ in the revision.
>
> ---
> [1] Simon, et al., arXiv:1905.00414
>
> [2] Nelson, et al., arXiv:2209.10652.
>
> [3] Skean et al., arXiv:2502.02013.
>
> [4] Minder et al., arXiv:2504.02922.
>
> [5] Wang et al., arXiv:2506.19823.
>
> [6] Blaine, et al., s41597-020-00715-8

---

> > ### Author Rebuttal · Reviewer_RvdD · 2026-04-04
> >
> > I appreciate the authors' detailed and constructive responses to my concerns. The authors have demonstrated the practical viability and cross-domain robustness of their interpretable framework. They have fully addressed my concerns.

---

> > > ### Author Response · Authors · 2026-04-07
> > >
> > > Thank you for the update and for confirming that your concerns are fully resolved. We truly appreciate your time and thoughtful feedback. If you feel it is appropriate, we would be grateful if the score could be adjusted to reflect these clarifications. We are happy to provide any further details if helpful.

---

### Official Review · Reviewer_fRuD · 2026-03-13

**Soundness:** 3
**Presentation:** 2
**Significance:** 3
**Originality:** 3
**Overall Recommendation:** 4
**Confidence:** 2

**Summary:**

The paper proposes a latent-level model diffing framework that interprets and compares internal representations of segmentation models across datasets, revealing dataset-specific features responsible for failures and identifying important internal features and deliberately adjusting them to improve segmentation without retraining.

**Compliance With Llm Reviewing Policy:**

Affirmed.

**Key Questions For Authors:**

a.	How is the proposed work is diffetent from the recent work on sparse autoencoder interpretability and crosscoder-based model diffing **[1]**.

b.	The analysis extracts activations from a single intermediate layer of the segmentation model. How sensitive are the results to the choice of layer her. For example, would earlier or later layers produce different latent structures or alignment results? Clarifying this would help determine whether the framework captures general representation differences or layer-specific artifacts.

c.	The paper shows that modifying specific latent features can improve segmentation results. In practical deployment scenarios, how would these corrective latents be identified without access to ground-truth labels or detailed failure analysis.

**[1]** Jiralerspong, T. and Bricken, T. Cross-architecture model diffing with crosscoders: Unsupervised discovery of differences between llms. In Mechanistic Interpretability Workshop at NeurIPS 2025.

**Limitations:**

yes

**Strengths And Weaknesses:**

a.	**Soundness:** The paper is generally technically sound and uses appropriate methods to analyze internal representations of segmentation models. The proposed framework combines tools such as sparse autoencoders, latent feature alignment, and intervention experiments to study representation differences across datasets. The empirical evaluation supports the main claims. However, the work is primarily empirical and lacks explicit theoretical justification. For example why sparse autoencoder features reliably correspond to meaningful segmentation concepts.

b.	**Presentation:** The paper is generally well written and easy to follow. However, some parts of the methodology could be improved by explicitly justifying, for example, why sparse autoencoder features correspond to meaningful segmentation concepts. Readers unfamiliar with interpretability literature may find some sections difficult to follow, and providing clearer explanations would further improve clarity.

c.	**Significance:** This work addresses an important problem, particularly in medical imaging, where models trained on one population or imaging protocol often perform poorly on others. By analyzing internal model representations, the work provides a useful framework for studying how segmentation models encode dataset-specific information. The ability to correct errors through latent feature interventions without retraining is also an interesting direction that could inspire future research.

d.	**Originality:** This work can be considered an original contribution. Although it combines existing components such as sparse autoencoder feature discovery and latent steering techniques, applying these methods to analyze segmentation models across datasets provides a new perspective. The latent-level model diffing framework and the demonstration that dataset-specific latent features influence segmentation performance offer useful insights into model behavior.

---

> ### Author Rebuttal · Authors · 2026-03-30
>
> We thank Reviewer fRuD for their thorough feedback. We address each point below.
>
> ---
> **a. Difference from prior [1].**
>
> Thank you for the insightful question. While our work is related to model diffing and sparse autoencoder interpretability, it differs from [1] in several key aspects.
>
> First, [1] focuses on language models and analyzes behavioral differences in generative settings, whereas we study dense prediction in medical image segmentation, where errors are spatial, clinically meaningful, and dataset shift plays a central role. In particular, we focus on settings where models are trained on the same anatomical task (e.g., tumor segmentation) but across different populations (e.g., pediatric vs. adult, or across geographic cohorts), and ask how these shifts are reflected in the internal latent structure (i.e., which features are shared versus population-specific).
>
> Second, [1] is primarily descriptive, identifying latent features that differ across models. In contrast, we show that these latents act as causal bottlenecks for segmentation failures and introduce targeted latent-level interventions that directly modify activations at inference time, enabling error correction and cross-dataset adaptation without retraining.
>
> Third, rather than only comparing models, we explicitly perform cross-dataset latent alignment to decompose representation differences into shared and dataset-specific components, allowing us to diagnose and mitigate failure modes induced by demographic and domain shifts.
>
> Finally, we validate this mechanistically by demonstrating that intervening on these latents leads to substantial performance recovery, moving beyond interpretation to actionable control.
>
> In summary, while [1] focuses on uncovering behavioral differences between models, our work provides a mechanistic, intervention-driven framework for understanding and correcting dataset shift in medical segmentation models.
>
> ---
> **b. Sensitivity to encoder layer choice.**
>
> Thank you for the insightful question. In the main paper, we use a mid-level layer motivated by prior work showing that intermediate layers capture the richest semantic structure [2, 3, 4]. We also include an ablation (Appendix B.1, Page 12) comparing early, mid, and late encoder layers.
>
> We observe systematic and interpretable differences across depth. Early layers produce diffuse, globally distributed latents that capture coarse anatomical context. Mid-level layers yield more localized and task-relevant features, balancing tumor and surrounding structure. Late layers produce highly concentrated, tumor-specific activations with lower entropy, as shown in Figure 9 on page 12 and Figure 10 on page 13.
>
> Importantly, these differences are consistent with standard hierarchical representations in deep models rather than artifacts of our method. While the exact latent structure varies with layer choice, the overall conclusions of our framework, such as identifying dataset-dependent latent usage and enabling targeted interventions, remain stable. We therefore use the mid-level layer as a principled trade-off between interpretability and task relevance.
>
> ---
> **c. How are corrective latents identified without labels.**
>
> Thank you for the insightful question. Ground-truth labels are only required during development to identify latents using the training data (e.g., via automated interpretation). At deployment time, no labels are needed. Once identified, corrective latents correspond to semantically meaningful features (e.g., edema-related or background suppression latents), and this mapping is fixed and reused across all experiments without re-identification. These latents can be monitored directly from the model’s activations, and interventions can be triggered based on activation patterns (e.g., unusually high or low latent responses).
>
> For example, in subsection (6.1. Instance-Level Failure Diagnosis), we identify the top-K most active SAE latents and intervene on them via latent steering, showing that interventions can be applied directly based on activation patterns at inference time. In subsection (6.2. Class-Level Performance Disparities), we focus on improving the edema class due to its lower Dice performance; the edema-specific latents used for intervention are pre-identified using automated interpretation on the training data and are reused during inference without requiring re-identification. Similarly, in our out-of-distribution experiments (Section 7), which simulate deployment, we identify domain-specific latents offline and then apply fixed intervention strategies at inference time on new data without access to ground truth.
>
> Overall, this demonstrates that labels are only needed once to discover and interpret corrective latents, after which latent steering can be applied as a lightweight, label-free adaptation mechanism at deployment.
>
> ---
>
> [2] Skean et al., arXiv:2502.02013.
>
> [3] Minder et al., arXiv:2504.02922.
>
> [4] Wang et al., arXiv:2506.19823.

---

> > ### Author Rebuttal · Reviewer_fRuD · 2026-04-03
> >
> > I appreciate the authors responses and they addressed my concerns adequately.

---

> > > ### Author Response · Authors · 2026-04-07
> > >
> > > Thank you for the update and for confirming that your concerns are fully resolved. We truly appreciate your time and thoughtful feedback. If you feel it is appropriate, we would be grateful if the score could be adjusted to reflect these clarifications. We are happy to provide any further details if helpful.

---

### Decision · Program_Chairs · 2026-04-30

**Decision:**

Accept (regular)

**Comment:**

This paper introduces a distinctive and well-motivated direction for interpretable medical image segmentation by combining sparse autoencoders, latent-level model diffing, cross-dataset latent alignment, and targeted latent interventions. The cross-architecture and cross-dataset analysis medically meaningful, especially in the context of population shift. The review discussion is clearly positive overall. The visible panel is 4/5/4/4, with no reject signal. One reviewer explicitly described the perspective as highly pioneering, and the visible rebuttal acknowledgements from three reviewers are fully resolved; the remaining reviewer also stays on the accept side. The authors are encouraged to incorporate their rebuttals into the final version of the paper. Overall, I recommend Accept.